# Developing Versatile Graphic Map Load Metrics

**Radek Barvir and Vit Vozenilek \***

Department of Geoinformatics, Palacky University Olomouc, 17. listopadu 50,
771 46 Olomouc, Czech Republic; radek.barvir@upol.cz

**\*** Correspondence: vit.vozenilek@upol.cz; Tel.: +420 585 634 513

**Abstract:** Graphic map load is a property of a map quantifying the amount of map content. It indicates the visual complexity of the map and helps cartographers to adapt maps and other geospatial visualizations to accomplish their purpose. Generally, map design needs to enable the user to quickly, comprehensively, and intuitively obtain the relevant spatial information from a map. Especially, this applies in cases like crisis management, immunology and military. However, there are no widely applicable metrics to assess the complexity of cartographic products. This paper evaluates seven simple metrics for graphic map load calculation based on image analytics using the set of 50 various maps on an easily understandable scale of 0–100%. The metrics are compared to values of user-perceived map load survey joined by 62 respondents. All the suggested metrics are designed for calculation with easy-accessible software and therefore suitable for use in any user environment. Metrics utilizing the principle of edge detection have been found suitable for a diversity of geospatial visualizations providing the best results among other metrics.

**Keywords:** map load; visual complexity; graphic load; metrics; map evaluation

## 1. Introduction

Map load, together with similar terms, e.g. map complexity or map density, is determined for quantifying the amount of map content. As Harrie et al. [1] consider, map complexity level can influence readability for users. Therefore, designing maps with the proper level of map load leading to adequate map complexity underlines its essential role for disciplines where effectivity of map-reading process is crucial, e.g. crisis management [2]. Even though the term "map complexity" is more frequently used when aiming for map readability evaluation, "map load" is being applied in this paper as the amount of graphics does not necessarily fully describe the complexity itself [3]. Inour study, graphic map load is represented by several indices quantifying image representations of maps influenced by map symbols' design and their distribution.

By the time when researchers have been focusing on map load, there has been a consensus on dividing it to intellectual (also called information) map load and graphic (visual) map load [4]. While the intellectual map load is influenced by both map reader knowledge and skills as well as by surrounding conditions, the graphic map load is more comfortable to be measured and compared between map samples [5]. Although the concepts of graphic map load vary author by author [4,6], this map property can be interpreted as a fullness of a map covered by map symbols and labels influenced by their spatial density, parameters (shape, size, fill etc.) and spatial distribution [7]. There are also researchers recognising more specific types of map load, e.g. [6,8,9]. For example, few studies [10,11] recognise label density. On the other hand, Robinson [5] and Brophy [12] discuss even the distinctiveness of graphic and intellectual map load. This paper aims to establish a metric for effective quantification of graphic map load suitable for a wide variety of map styles.

Map load concerning its equivalent terms have been studied by cartographers since the middle of the 20th century [5] and peaked in the 1970s and 1980s by studies of Sukhov [13,14] and others. In

the early stages of that research, there were many various approaches to examine map load through vector measurements based on the graph theory using vertex and edges count and length [6] or map symbols area coverage [10]. Those theoretical principles were though applicable only for specific cartographic methods, e.g., choropleth and isopleth maps. They were very limited or even impossible to be used for a wide range of other map types, as later discussed by Fairbairn [8].

Later, Shannon's Entropy became the advanced approach applied especially for intellectual map load analysis. There were several studies based on the idea that a higher level of uncertainty with more unexpected information leads to more complex maps [15]. Entropy in terms of map load measurements and map generalization was deeply examined by Bjørke [16], Sukhov [13,14] and Neumann [17].

While many theoretical approaches enriched the concept of map load measuring, empirical experiments on map complexity have occurred much later and still lack [18]. Conversely, Brandli [19] examined unconventional methods for map load estimation by user tasks evaluation. Additionally, eye-tracking techniques [20–23] and compression rate [24] contributed in challenging research on map load. Unfortunately, those methods based on user testing cannot provide results comparable with results obtained at different time with another respondent group with different knowledge and skills.

In recent years, the attention of researchers moved from intellectual to visual complexity as a more objective and comparable map property [18] and from vector to raster representations due to large scope of vector formats [25]. Ai et al. [26] recently presented a very simple metric using the count of specific pixels representing each symbol as a simple metric to estimate the map load of ocean flows. Analysing maps in an image file format also provide a method to evaluate such complex visualizations as topographic maps and city plans are [27]. In recent years, image processing aiming for map load measures became a primary way of map complexity estimation [18,24,28].

Moreover, measuring map load is significant not only in the mapmaking process. It may also be used to evaluate and compare existing maps and map collections to find differences between map producers, styles and different cultures of origin. There are large collections of both contemporary and historical maps, e.g., The United States Geological Survey (USGS) topographic map archive, UK Ordnance Survey, Swiss Siegfried maps, and Sanborn fire insurance map archive, etc. Those sources could be examined using spatiotemporal analysis, data mining and image analyses to retrieve map load values in a wider scope to reveal changing trends in cartography in terms of map loading and complexity [29]. Advanced approaches using neural networks for comprehensive map archives were studied by Petitpierre [30].

## 2. Materials and Methods

Based on current trends and authors' ideas, they proposed three approaches to measure the graphic map load using raster map formats: average darkness (AD), image compression (IC) and edge detection (ED). All three approaches meet predefined conditions:

- Are based on processing image representation of maps;
- Are simple to apply widely;
- Are measurable on an accessible platform in a short time;
- Are applicable for a wide range of map styles;
- Can be measured on a clearly delimited range of values.

The first approach average darkness (AD) evolves the idea that the darker pixels occur in a map image relatively to empty bright pixels, the more loaded the map is. The image compression (IC) approach builds the on comparison of file sizes of uncompressed and compressed image files. In contrast, the third approach edge detection (ED) uses an edge detection filter to evaluate the presence of both hard and soft edges in a map image. The authors developed several metrics for these approaches and described them in Sections 2.2–2.4. All the metrics were performed in IrfanView 4.52 software using built-in tools.

*2.1. Reference Map Set and Its Evaluation of User-Perceived Map Load*

As no standard way to measure graphic map load exists, a user experiment was prepared to gain a set of user-perceived values of the graphic map load for comparing with the developed metrics based on digital image analyses. Firstly, a set of 50 various maps was collected for providing a reference set (Ref-Set) for both measured and perceived map load. The maps differed in their topic (topographic, geographic, thematic), purpose, map scale and size, aspect ratio, area covered, map style, level of generalization and age of origin. The set contained cadastral maps, military maps, maps with orthophoto base layer, old maps, urban plans, transport schemes, other thematic maps and sections of popular web map portals. The scope of diversity was considered wider than in other similar studies, and therefore sufficient to fulfil the goal of validating the metrics. All maps were obtained in a digital image format even though some of them were designed to be paper maps. From the total number of 50 maps, 18 maps were cropped from web-map applications, 22 were exported from a graphic or GIS software and 10 maps were scanned. All the digital images of the maps were collected and stored in the 8-bit depth for each of the red, green and blue (RGB) channels resulting in 24 bit per pixel (BPP) in total, which is the most common structure of storing digital images.

Further, testing of user-perceived map load (UP) followed up to obtain a reference sample of map load values. Then the sample was used to be compared with metrics' measurements. The sample of 62 respondents joined the experiment comprising both people with various levels of cartographic education (ranging from first-year students of cartography up to experienced cartographers) and people without previous cartographic knowledge nor skills. There were 22 respondents with no previous cartographic education (36% of the total number), 17 graduates of a five-month-long basic course of cartography and geographic information systems (GIS) (27%) and 23 well-experienced ones designing maps regularly on a weekly basis at the time of the evaluation (37%).

The experiment always began with a brief introductory into graphic map load meaning presented to the users. Then, a subset of twenty-five maps from the Ref-Set was displayed for two seconds each to illustrate the scope of map load in the Ref-Set. The reason was to prevent the respondents from the relative comparison of maps in subsequent evaluation. After this freewatching, each individual respondent watched each of the 25 maps for 18 seconds and during this time they evaluated the graphic map load of each map shown on the stimulus (slide showing a map). They wrote down their subjective perceived value of graphic map load in a scale ranging from 0 to 100% (stored as value $L_{UP} \in <0;100\%>$). Moreover, they subjectively estimated if the map was either subloaded, overloaded or loaded adequately to the expected map purpose by ticking one of those possibilities. These subjective evaluations were stored as $L_{SE} \in \{subloaded; loaded\ adequately; overloaded\}$. The time of 18 seconds was set after the consideration that around 6 seconds are necessary to watch the map layout and evaluate the visualisation methods used, another 6 seconds to process the ideas (to think about the percentage of map load) and another 6 seconds to write the values down and get ready for another stimulus. This assumption was also examined in a small pre-testing joined by 5 respondents, all indicating this time was sufficient to fulfil the task. The count of 25 of the 50 maps per respondent was chosen to prevent the respondents from getting bored and inattentive by the long duration of the experiment and so to avoid receiving inaccurate answers in the latter part of the experiment. The experiment composed of a short introduction, task description and both the freewatching and the evaluating part took usually 12 to 15 minutes, which fits the attention span described commonly in psychological research [31,32].

Each map was displayed in full extent and a detailed cutout in this part of the experiment to cover both global and accurate way of map perception (Figure 1). The total number of 62 sets of records of $L_{UP}$ values and subjective evaluations $L_{SE}$ of graphic map load adequacy was collected. Finally, few details about the respondents themselves were collected. Respondents recorded their evaluations into pre-printed paper forms, which were then aggregated into a digital file for further processing.

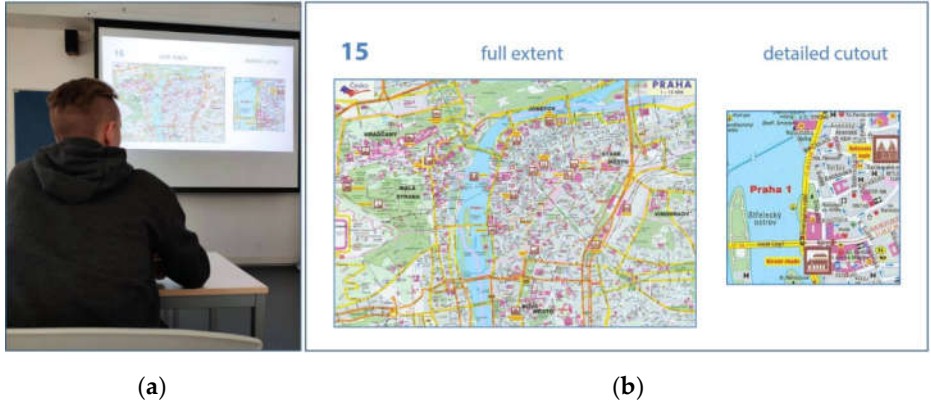

(**a**)          (**b**)

**Figure 1.** User experiment in progress (**a**) with an example of map stimuli in its second phase (**b**).

The same Ref-Set, whose map load was first subjectively evaluated by respondents and marked as L$_{UP}$, was also measured using metrics based on image analytics described in Sections 2.2–2.4. Values provided by the metrics were then compared with the average user-perceived values using Pearson correlation coefficient followed by calculating t-statistics and *p*-value, and Euclidean distance. The L$_{UP}$ values categorised to three populations according to L$_{SE}$ were also evaluated using Kruskal–Wallis nonparametric test.

## 2.2. Measuring Map Load Using Average Darkness Approach

The AD approach was based on the average pixel darkness of a map (Figure 2). The metric principle of AD comes from an assumption: the darker map, the more graphically loaded the map. The AD approach considers that maps mostly have a bright background while the map symbols are performed in darker colours. Therefore, the more content there is present in a map, the darker the map is supposed to be. In contrary, for mostly empty maps, the light background is expected to cover most of the map area and the average darkness is lower. This approach evolves the old principle of map load calculation [10] used mostly for topographic maps with a standardised map key. Some up-to-date thematic maps, though, use various graphic styles and colour schemes including inverse-design principle which is expected to be problematic for map load evaluation. Even though, still the majority of maps including the thematic ones use the principle the more intensive phenomenon the darker (more intensive) colour. The AD metric's principle could be likened to how much ink would be necessary to print the map in black and white relatively to how much ink would be necessary to print the black area of the same size.

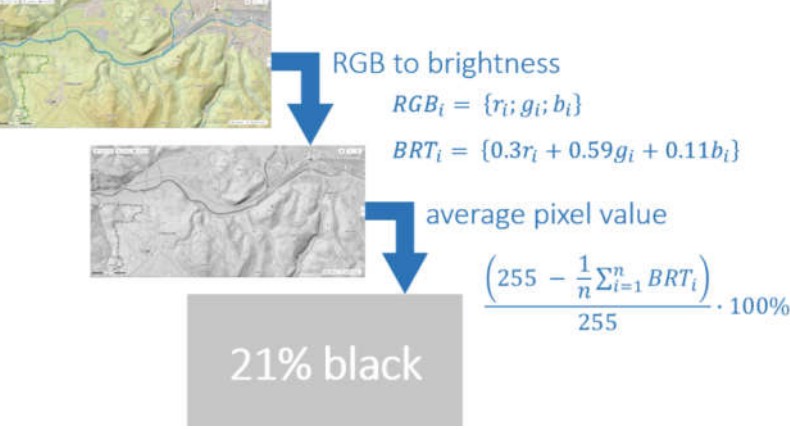

RGB to brightness

$$RGB_i = \{r_i; g_i; b_i\}$$

$$BRT_i = \{0.3r_i + 0.59g_i + 0.11b_i\}$$

average pixel value

$$\frac{\left(255 - \frac{1}{n}\sum_{i=1}^{n} BRT_i\right)}{255} \cdot 100\%$$

21% black

**Figure 2.** Principle of the average darkness (AD) metric.

Each image file standing for a map from the Ref-Set was processed in IrfanView 4.52 software to obtain values of graphic map load according to AD metric. The average pixel brightness was computed by the histogram tool. The software calculated the brightness of each pixel as 0.299R + 0.587G + 0.114B (representing multiple of each RGB channel value). The received number was on a scale 0–255 where 0 represented black and 255 white colours. The L$_{AD}$ (representing the graphic map load according to AD metric) value needed to be transformed Equation (1):

$$L_{AD} = \frac{100\%}{255} \cdot (255 - BRT),$$  (1)

where: *BRT* represents the average pixel brightness index displayed in IrfanView.

### 2.3. Measuring Map Load Using Image Compression Approach

Another approach for graphic map load measurements was developed considering the difference in compression effectivity. The IC metrics used a ratio of compressed to uncompressed image file size (in kB) representing each map from the Ref-Set. Various image file formats and their compressed forms were examined, resulting in metrics IC1–IC3. The IC1 metric calculates a ratio of the size of slightly compressed JPG file (quality 90/100) to the size of uncompressed TIF file, while the IC2 metric was using highly compressed JPG file (quality 20/100) in the numerator instead. The IC3 metric quantified the size ratio of TIF compressed by LZW algorithm to uncompressed TIF file. All image format transformations were processed in IrfanView and map load values L$_{IC1}$, L$_{IC2}$ and L$_{IC3}$ were calculated Equations (2)–(4):

$$L_{IC1} = \frac{s_{JPG90}}{s_{TIF}},$$  (2)

$$L_{IC2} = \frac{s_{JPG20}}{s_{TIF}},$$  (3)

$$L_{IC3} = \frac{s_{TIFLZW}}{s_{TIF}},$$  (4)

where:

- *s* represents respective image file size in kB
- *JPG90* is a map image stored in JPG format with quality 90/100, *JPG20* with quality 20/100
- *TIFLZW* is a map image stored in TIF format with LZW compression
- *TIF* represents a map image stored in uncompressed TIF file format

### 2.4. Measuring Map Load Using Edge Detection Approach

The edge detection (ED) approach uses a filter to detect edges in map images. In this experiment, a fuzzy approach was used taking into account sharp as well as soft edges, unlike in some related articles, e.g. [18], with applying binary principle distinguishing edge and non-edge pixels. The principle of the ED metric is that the more and the sharper edges in a map, the higher graphic map load it is. While large areas covered by the same colour make the map simple, the colour transitions represented by neighbouring map symbols make the map more complex. Some researchers [18,27,33] also describe this idea. Three metrics ED1–ED3 were designed. The first step in all of them consisted of measuring the map load value L$_{ED}$ using a built-in filter to convert each map image into an edge detection image. Brighter colours in the edge detection image indicated sharper edges (Figure 3).

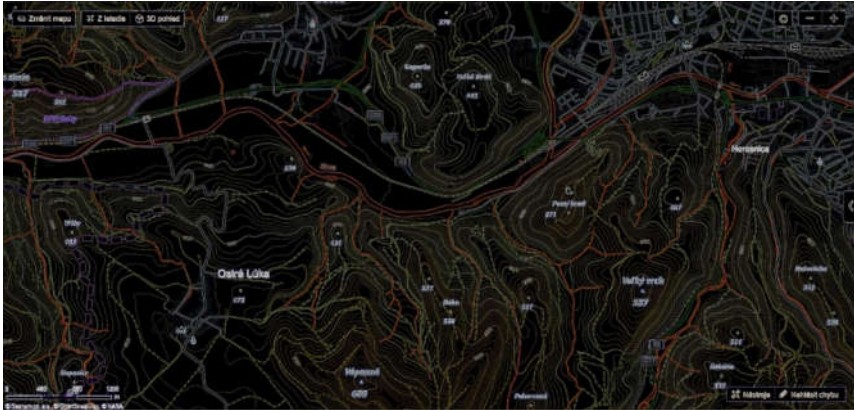

**Figure 3.** Edge detection image of one of the maps from the Ref-Set.

The edge detection image was then processed similarly to the AD metric (see Sections 2.2) by investigating the average brightness of average pixel (Figure 4). As bright pixels represented more loaded parts of a map and dark ones the less loaded parts, the average pixel values were just shrunk to scale 0–100% instead of 0–255. The calculation process of map load $L_{ED1}$ is presented in Equation (5):

$$L_{ED1} = \frac{100\%}{255} \cdot EDD,$$ (5)

where:

- *EDD* represents the average pixel brightness of the edge detection image displayed in IrfanView

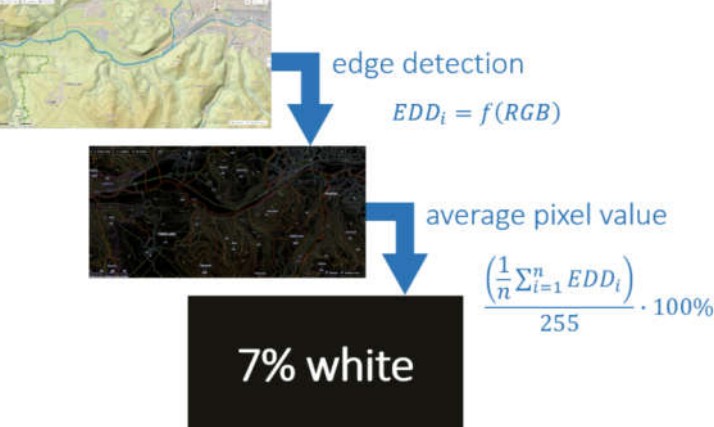

**Figure 4.** Principle of the ED1 metric.

Because the map load values $L_{ED1}$ for most maps in the Ref-Set ranged only a few percent, the metrics ED2 and ED3 were developed to use a wider scope of 0–100% scale rather than just its lower part while still preserving 0 and 100% thresholds. Map load values $L_{ED2}$ and $L_{ED3}$ were therefore derived by rooting $L_{ED1}$ value according to Equations (6) and (7):

$$L_{ED2} = \sqrt{L_{ED1}},$$ (6)

$$L_{ED3} = \sqrt[3]{L_{ED1}},$$ (7)

*2.5. Bit-Depth Experiment*

A small experiment was also performed to evaluate how using various bit depths could potentially influence the results provided by the metrics. As all maps from the Ref-Set were obtained in the RGB colour space with 24 BPP, two maps M4 and M32 were also converted to 8 BPP and 4 BPP. This was done using the decrease color depth tool in IrfanView with the Floyd–Steinberg dithering settings. While M4 presented a map cropped from a web map application and had never been printed on a paper before, M32 had been digitised by scanning. Both the maps in all three variants (24, 8 and 4 BPP) were then measured by all presented metrics in the same way as in Sections 2.2–2.4.

## 3. Results

*3.1. User-Perceived Map Load Evaluation*

Data obtained during the initial experiment were first digitised into an electronic sheet for further processing. The scope of 0–100% scale differed from one respondent to another. While some respondents perceived the average map load to be only 22%, the respondent with the highest average $L_{UP}$ value reached 80%. It was caused by giving the respondents a considerable degree of freedom, which limited only the minimum and maximum values. Because of that, they could express their personal feeling of map load. Subsequently, the data for each respondent had to be normalized to the range 0–1 according to the minimum and maximum values for later correlation measures.

However, obtaining exact values of graphic map load of each map from the Ref-Set was just a minor part of the study. An essential task was to get a view which maps from the Ref-Set are the more loaded ones and which, on the other hand, are the less graphically loaded and how much. Letter M and a number marked each of the 50 maps, so there were maps M01–M50. For six samples from the Ref-Set, maps M5, M16, M27, M29, M40 and M49 (some shown in Figure 5), the most frequent evaluation of $L_{SE}$ was that they were subloaded to expected map purpose. In contrary, M15, M17, M20, M25, M32, M35, M39, M41, M45, M47 and M48 were mostly evaluated to be overloaded (some shown in Figure 6). The rest of the maps from the Ref-Set were mostly marked as loaded adequately.

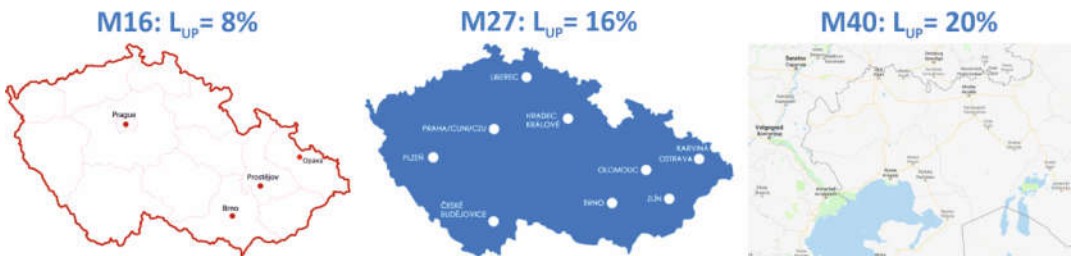

**Figure 5.** Maps with the lowest $L_{UP}$ values also marked by respondents as subloaded in $L_{SE}$ (two simple maps of Czechia and cutout of Google maps focused on the south part of Kazakhstan).

The highest average user-perceived map load values were detected for M45 ($L_{UP}$ = 88%), M39 and M48 (both $L_{UP}$ = 82%). Four more maps exceeded LUP of 80%. The significantly lowest value of user-perceived graphic map load, on the other hand, had M16 ($L_{UP}$ = 8%). The second less-loaded was M27 ($L_{UP}$ = 16%) followed by M40 ($L_{UP}$ = 20%) as captured in Figure 5.

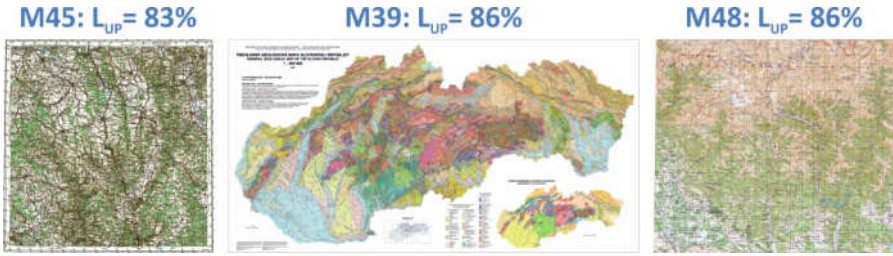

**Figure 6.** Maps with the highest L$_{UP}$ values also marked by respondents as overloaded in L$_{SE}$ (two sections of topographic maps and a geological map of Slovakia).

As a part of the user-experiment evaluation, both L$_{UP}$ and L$_{SE}$ were visualized together in Figure 7. For each respondent R1–R62, L$_{UP}$ were coloured according to L$_{SE}$ (green representing map load values for maps evaluated as subloaded, yellow for maps loaded adequately to their map aim and red representing overloaded ones). This visualisation captures how subjective evaluation can use a wide range of scopes and differs one from another respondent and, therefore, how an objective metric for map comparison is necessary. Dotted lines represent the average L$_{UP}$ for each of the L$_{SE}$ classes in representative colour.

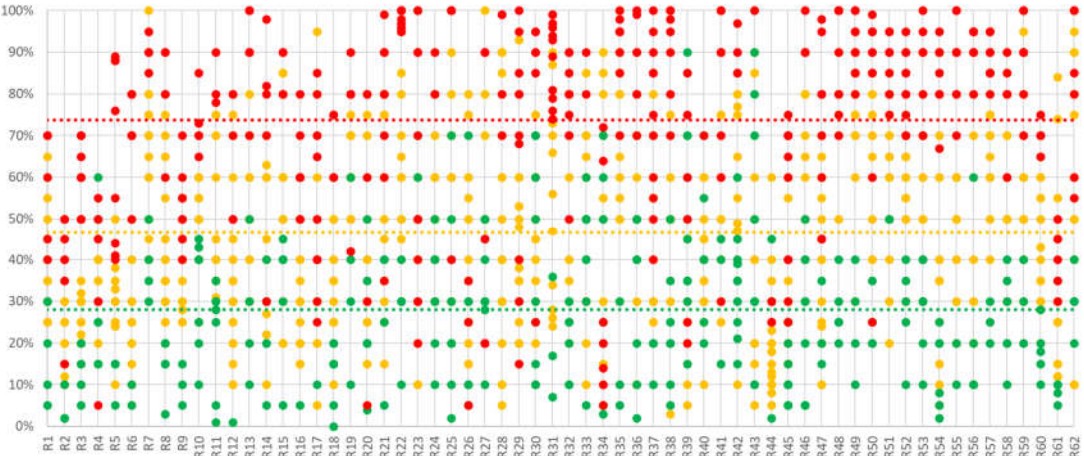

**Figure 7.** Respondents' evaluation of L$_{UP}$ of the maps from the Ref-Set coloured by L$_{SE}$ (green representing subloaded maps, red overloaded and yellow maps loaded adequately to their aim).

For the Kruskal–Wallis test, the null hypothesis was set saying all three populations (subloaded, overloaded, loaded adequately) contain the same L$_{UP}$ values. The sampling distribution vas calculated to be H = 720,08. This value was then compared to $\chi^2$a,k-1 reaching 5.99 (less than 720,08) and so, according to *p*-value $4.33 \cdot 10^{-157}$, the null hypothesis was rejected. This means the populations categorised according to L$_{SE}$ reached different levels of user-perceived map load (L$_{UP}$).

*3.2. Average Darkness Approach*

The L$_{AD}$ values measured in IrfanView software were added to the summary table along with L$_{UP}$ values. Pearson correlation coefficient and Euclidean distances between average L$_{UP}$ and L$_{AD}$ were then calculated for each map from the ref-set. The results reveal no significant link between L$_{AD}$ (orange points) and L$_{UP}$ (blue points) values (Figure 8). While L$_{AD}$ for most topographic and some thematic maps with white background rose adequately to L$_{UP}$, for maps with a coloured and black background, e.g., M13 and M24 (Figure 9), the AD metric failed due to ignoring the trend of user-perceived map load. In addition, maps with orthophoto background, e.g., M7, M8, M10, were evaluated with much higher values than other maps, as visible in Figure 8. The correlation coefficient 0.07 indicated no significant correlation between L$_{UP}$ and L$_{AD}$ value sets, which was also supported by *p*-value of t-statistics reaching 0,64. Euclidean distance value was 1.62. The result of the Pearson correlation indicated that the AD metric is not suitable for graphic map load evaluation.

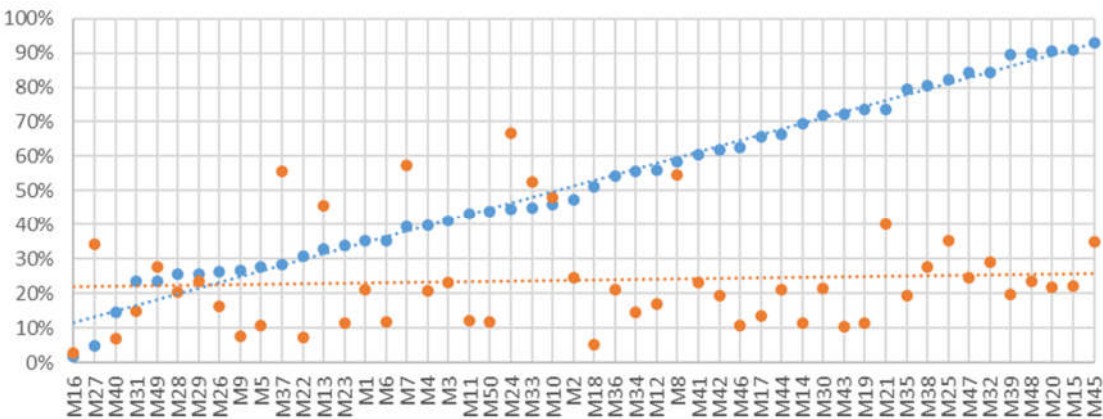

**Figure 8.** Graphic map load of the Ref-Set maps from user testing ($L_{UP}$, marked blue) and measured by the AD metric ($L_{AD}$, marked orange) sorted in ascending order by $L_{UP}$, dotted lines indicate trend.

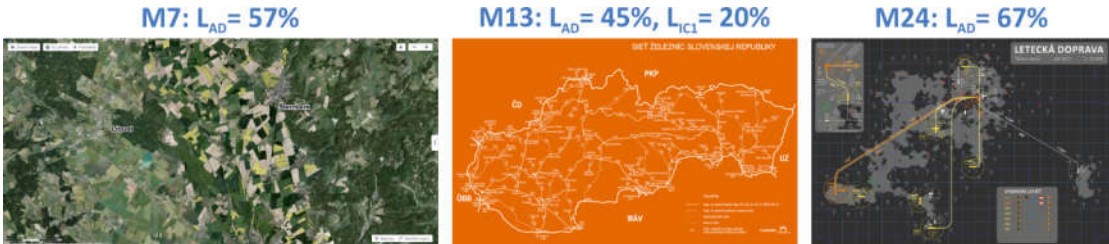

**Figure 9.** Examples of the maps where map load values were overestimated using AD metric.

### 3.3. Image Compression Approach

Similarly, IC1, IC2 and IC3 metrics through respective graphic map load values $L_{IC1}$, $L_{IC2}$ and $L_{IC3}$ were statistically compared with $L_{UP}$ values. For the IC1 metric using JPG image compression with 90% image quality, the correlation coefficient of 0.55 ($p$-value $3.41 \cdot 10^{-5}$) and Euclidean distance 2.01 were calculated. In the case of the IC2 metric, the indicators showed a slightly better correlation comparing to the ED1 metric resulting in 0.60 ($p$-value $4.88 \cdot 10^{-6}$) and Euclidean distance 1.97. Finally, when examining IC3 metric, Pearson correlation of 0.56 was detected ($p$-value $2.87 \cdot 10^{-5}$), and the Euclidean distance 1.32 fits better with the scope used by respondents. According to the $p$-values, a statistically significant correlation was found between sets of $L_{UP}$ and $L_{IC1,2,3}$.

Even though the IC metrics provided generally better results than AD metric, still several maps from the Ref-Set were evaluated with the intense disorder to user perception occurred. In the case of the IC1 and IC2 metrics based on JPG compression, samples M13, M20, M38, M45 and M47 became a sort of outliers with extremely high values comparing to the trend of other maps (Figure 10). Those maps represent mostly complex topographic with lots of symbols, several different colours used and white background, and contrast old maps (Figure 11), but also a quite simple monochromatic map M13 (Figure 9).

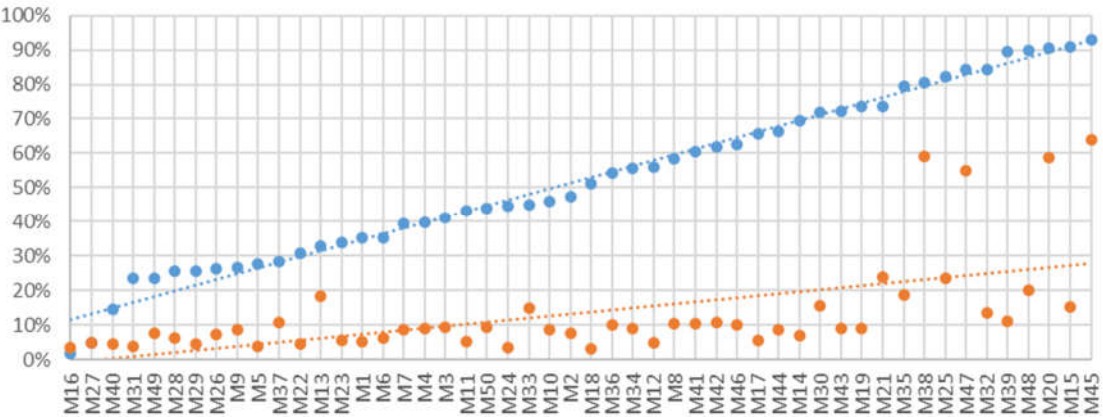

**Figure 10.** Graphic map load of maps from the Ref-Set obtained during user testing ($L_{UP}$, marked blue) and measured by IC2 metric ($L_{IC2}$, marked orange) sorted in ascending order by $L_{UP}$, dotted lines indicate trends.

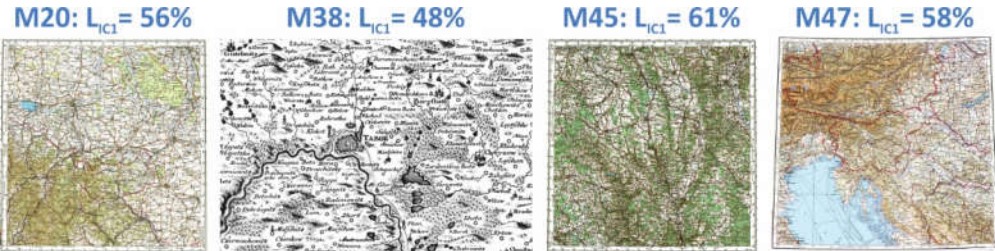

**Figure 11.** Examples of the maps where map load values were overestimated using IC1 metric.

In contrary, IC3 metric tends to be more chaotic and under-evaluating several maps from the Ref-Set. As captured in Figure 12, especially maps M11, M14, M16, M17, M22 and M24, all with a limited number of colours, obtained extremely low $L_{IC3}$ values. Therefore, according to the issues described above, the metrics based on the IC approach were found to correlate with user perception of map load. However, these metrics can be influenced by image characteristics and so may deviate from user-perceived values.

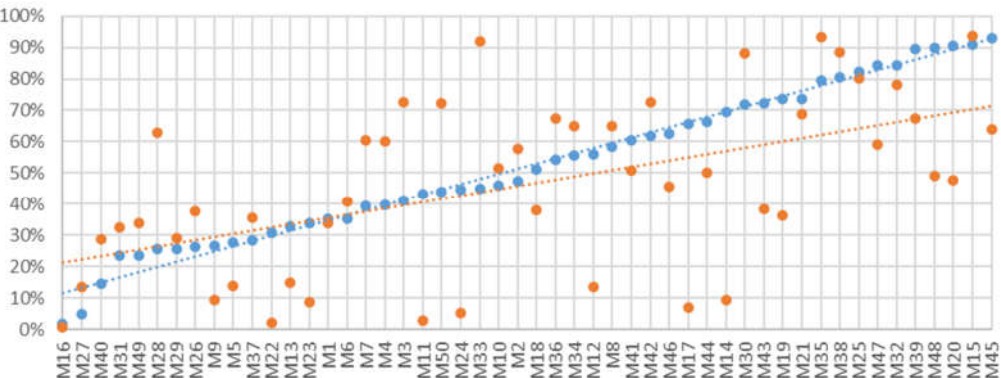

**Figure 12.** Graphic map load of maps from the Ref-Set obtained during user testing ($L_{UP}$, marked blue) and measured by IC3 metric ($L_{IC3}$, marked orange) sorted in ascending order by $L_{UP}$, dotted lines indicate trends.

### 3.4. Edge Detection Approach

The results for all ED1, ED2 and ED3 metrics brought higher correlation coefficients with average values of $L_{UP}$. In the case of ED1, it was 0.72 ($p$-value $2.73 \cdot 10^{-9}$), for ED2 even 0.77 ($p$-value $9.17 \cdot 10^{-11}$) and for ED3 correlation of 0.77 ($p$-value $4.52 \cdot 10^{-5}$) was detected. Therefore, a statistically significant correlation was found especially between ED2 and UP, ED3 and UP metrics, respectively. As expected, the Euclidean distance declined with the rising ED metric index—2.05 for ED1, 1.21 for ED2 and 0.69 for ED3 metric.

Figure 13 shows a higher level of harmony between $L_{ED3}$ and $L_{UP}$ values comparing to $L_{ED1}$, especially in the first half of the map samples ordered ascending by the average user-perceived graphic map load. In the second half, deviations rise. However, no extreme disorders were found between the values measured using the ED3 metric and the respondents' estimations. Higher differences were detected for the samples M14, M17, M19 and M24. Those maps may seem complex at the initial sight, but can be easily interpreted when watched longer. Therefore, the limited time the respondents had for evaluation of the maps might have played some significant role in this issue.

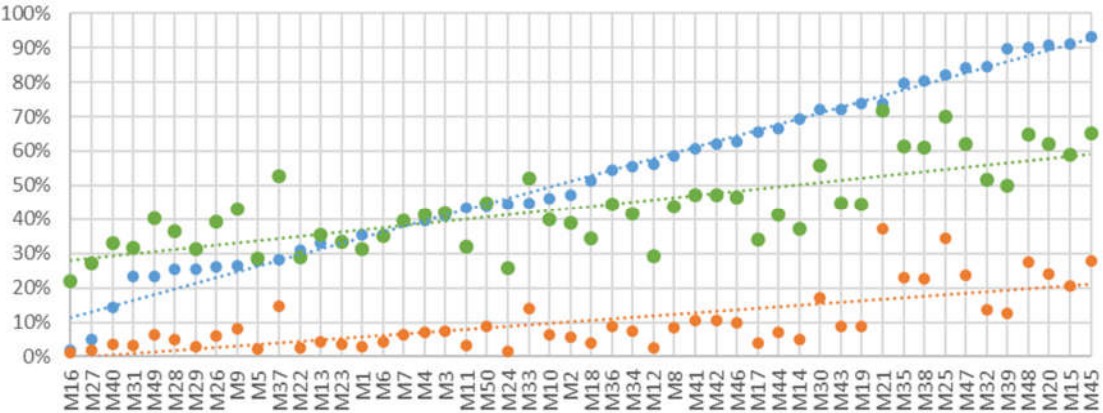

**Figure 13.** User-perceived values ($L_{UP}$, marked blue), map load values measured by ED1 metric ($L_{ED1}$, marked orange) and ED3 metric ($L_{ED3}$, marked green) sorted in ascending order by $L_{UP}$. Dotted lines indicate trends.

Map M21 (Figure 14) was the most graphically loaded according to all ED metrics, while according to UP, this map was the fifth most loaded one. This map represents an old army map with hatches used to express terrain. The edge detection filter interpreted the numerous dark hatches on its white background as sharp edges, and so contributed to higher $L_{ED}$ value. On the other hand, the values $L_{ED2} = 61\%$ and especially $L_{ED3} = 72\%$ were very close to reference $L_{UP} = 74\%$. The similar issue caused by the number of edges is evident on the old map of Silesia M25 (Figure 14). In addition, M37 (Figure 14) representing a grid map of the Minecraft world obtained higher values comparing to neighbouring maps sorted by $L_{UP}$.

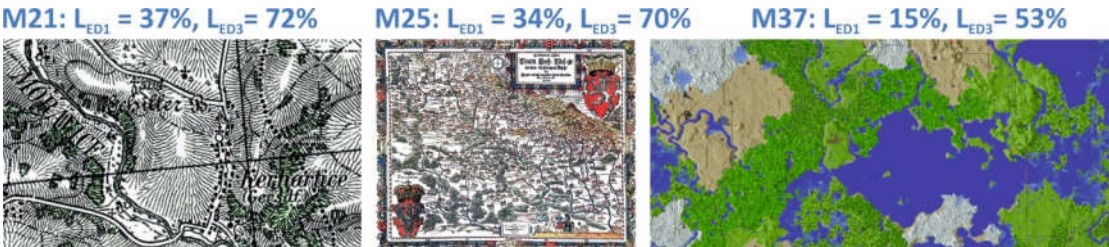

**Figure 14.** Maps M21 and M25 with the highest $L_{ED}$ values and M37.

Table 1 shows the average normalised $L_{UP}$ values along with measured map load values for maps M01–M50. All numbers are rounded to integer, but for calculations, their precise values were used.

**Table 1.** User perceived and measured map-load values of maps from the Ref-Set [%].

| Map | $L_{UP}$ | $L_{AD}$ | $L_{IC1}$ | $L_{IC2}$ | $L_{IC3}$ | $L_{ED1}$ | $L_{ED2}$ | $L_{ED3}$ |
|-----|------|------|------|------|------|------|------|------|
| M01 | 35 | 21 | 6 | 5 | 34 | 3 | 17 | 31 |
| M02 | 47 | 24 | 10 | 8 | 58 | 6 | 24 | 39 |
| M03 | 41 | 23 | 11 | 9 | 72 | 7 | 27 | 42 |
| M04 | 40 | 21 | 10 | 9 | 60 | 7 | 26 | 41 |
| M05 | 28 | 11 | 5 | 4 | 14 | 2 | 15 | 29 |
| M06 | 35 | 12 | 9 | 6 | 41 | 4 | 21 | 35 |
| M07 | 39 | 57 | 10 | 9 | 60 | 6 | 25 | 40 |
| M08 | 58 | 55 | 11 | 10 | 65 | 8 | 29 | 44 |
| M09 | 26 | 8 | 6 | 9 | 9 | 8 | 28 | 43 |
| M10 | 46 | 48 | 10 | 9 | 51 | 6 | 25 | 40 |
| M11 | 43 | 12 | 4 | 5 | 3 | 3 | 18 | 32 |
| M12 | 56 | 17 | 5 | 5 | 13 | 3 | 16 | 29 |
| M13 | 33 | 45 | 20 | 18 | 15 | 4 | 21 | 35 |
| M14 | 69 | 12 | 5 | 7 | 9 | 5 | 23 | 37 |
| M15 | 91 | 22 | 12 | 15 | 94 | 21 | 45 | 59 |
| M16 | 2 | 3 | 3 | 3 | 1 | 1 | 10 | 22 |
| M17 | 66 | 14 | 6 | 5 | 7 | 4 | 20 | 34 |
| M18 | 51 | 5 | 5 | 3 | 38 | 4 | 20 | 34 |
| M19 | 74 | 12 | 8 | 9 | 36 | 9 | 30 | 44 |
| M20 | 91 | 22 | 56 | 59 | 48 | 24 | 49 | 62 |
| M21 | 74 | 40 | 18 | 24 | 69 | 37 | 61 | 72 |
| M22 | 31 | 7 | 3 | 4 | 2 | 2 | 16 | 29 |
| M23 | 34 | 11 | 5 | 5 | 9 | 4 | 19 | 33 |
| M24 | 44 | 67 | 3 | 3 | 5 | 2 | 13 | 26 |
| M25 | 82 | 35 | 14 | 23 | 80 | 34 | 59 | 70 |
| M26 | 26 | 16 | 6 | 7 | 38 | 6 | 24 | 39 |
| M27 | 5 | 34 | 4 | 5 | 13 | 2 | 14 | 27 |
| M28 | 25 | 20 | 8 | 6 | 63 | 5 | 22 | 36 |
| M29 | 25 | 23 | 4 | 4 | 29 | 3 | 17 | 31 |
| M30 | 72 | 22 | 12 | 15 | 88 | 17 | 42 | 56 |
| M31 | 23 | 15 | 5 | 4 | 32 | 3 | 18 | 32 |
| M32 | 84 | 29 | 10 | 13 | 78 | 14 | 37 | 52 |
| M33 | 45 | 53 | 16 | 15 | 92 | 14 | 38 | 52 |
| M34 | 56 | 15 | 10 | 9 | 65 | 7 | 27 | 42 |
| M35 | 80 | 19 | 17 | 19 | 93 | 23 | 48 | 61 |
| M36 | 54 | 21 | 11 | 10 | 67 | 9 | 30 | 44 |
| M37 | 28 | 55 | 15 | 11 | 36 | 15 | 38 | 53 |
| M38 | 80 | 28 | 48 | 59 | 88 | 23 | 48 | 61 |
| M39 | 90 | 20 | 12 | 11 | 67 | 12 | 35 | 50 |
| M40 | 14 | 7 | 5 | 5 | 29 | 4 | 19 | 33 |
| M41 | 61 | 23 | 10 | 10 | 51 | 10 | 32 | 47 |
| M42 | 62 | 19 | 12 | 11 | 73 | 10 | 32 | 47 |
| M43 | 72 | 10 | 7 | 9 | 38 | 9 | 30 | 45 |
| M44 | 66 | 21 | 6 | 9 | 50 | 7 | 27 | 41 |
| M45 | 93 | 35 | 61 | 64 | 64 | 28 | 53 | 65 |
| M46 | 63 | 11 | 13 | 10 | 46 | 10 | 32 | 46 |
| M47 | 84 | 25 | 58 | 55 | 59 | 24 | 49 | 62 |
| M48 | 90 | 23 | 20 | 20 | 49 | 27 | 52 | 65 |
| M49 | 23 | 28 | 7 | 8 | 34 | 6 | 25 | 40 |
| M50 | 44 | 12 | 12 | 9 | 72 | 9 | 30 | 45 |

*3.5. Bit-Depth Influence*

Additionally, the results of the map samples M4 and M32 with lowered bit depth were collected and examined to discover how individual metrics are sensitive to this parameter. While the 24 BPP version offers 65,536 possible colours, the 8 BPP lower to 256 colours and 4 BPP to only 16 colours. This strong colour reduction, especially in the case of 4 BPP version, led to a dotted texture on monochrome areas (Figure 15) resulting in sharp changes of colour hue. While the AD approach was not influenced by this reduction much, in the case of ED approach, the lower bit depth resulted in a bit higher map load values. The most affected was the IC approach where decreasing the colour depth lowered significantly the file size of the uncompressed image files and, therefore, raised the final map load value. The map load values are noted in Table 2.

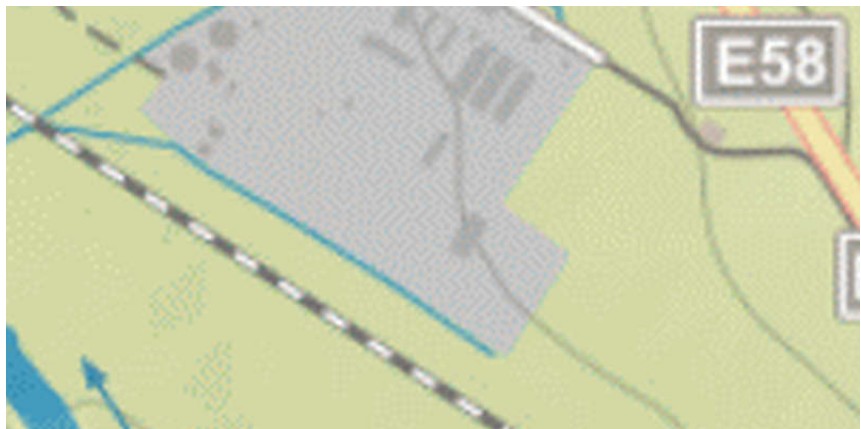

**Figure 15.** Dotted structures caused by colour depth reduction in the case of M4.

**Table 2.** Map load values (in %) of M4 and M32 with decreased bit depth.

| Map Load Metrics [%] | M4 24 BPP | M4 8 BPP | M4 4 BPP | M32 24 BPP | M32 8 BPP | M32 4 BPP |
|---|---|---|---|---|---|---|
| $L_{AD}$ | 21 | 21 | 21 | 29 | 29 | 31 |
| $L_{IC1}$ | 10 | 34 | 87 | 10 | 40 | 94 |
| $L_{IC2}$ | 9 | 27 | 54 | 13 | 41 | 83 |
| $L_{IC3}$ | 62 | 65 | 56 | 78 | 56 | 46 |
| $L_{ED1}$ | 7 | 8 | 12 | 14 | 15 | 17 |
| $L_{ED2}$ | 26 | 29 | 34 | 37 | 38 | 41 |
| $L_{ED3}$ | 41 | 44 | 49 | 52 | 53 | 55 |

As visible from Table 2, for the AD metrics, the map load values remained the same in the case of M4 for all bit depths and only increased from 29% to 31% in the case of the 4 BPP version of M32. For IC1 and IC2 metrics, the map load values are increasing significantly with lowering the bit depth. In the case of IC3 metric, for M4 the map load is the highest for 8 BPP and the lowest for 4 BPP, while for M32 is decreasing with lowering bit depth. For ED1, ED2 and ED3 metrics, the map load values are slightly increasing when lowering the bit depth. Therefore, the ED approach seems to be less dependent on the bit depth of images and even though the correlation was proved for both approaches, the ED approach tends to provide more consistent values and is more suitable to work as a versatile map load metric.

## 4. Discussion

As there has been no standard nor conventional way for graphic map load measurements available, the developed approaches with metrics were compared to the user-perceived evaluation experiment. Even in this group of respondents, the various scope of the 0–100% scale was used. Therefore, normalization of the data collection was done so any substantial influence of the scope

used seems to be present. Conversely, the user-perceived values were evaluated according to subjective feelings with no need of cartographic skills and therefore are expected to be similar to potential different user groups. As the aim of this study was to evaluate the potential of the new metrics rather than aiming for measurements of precise map load values of each map, the number of 62 respondents was considered sufficient for this purpose and comparable with other studies on map evaluation [22,34–36].

Additionally, the Ref-Set consisted of a limited amount of 50 various sample maps. The aim when compiling the Ref-Set was to choose various map styles differing in as many aspects as possible. Most of the maps originated in Czechia and had labels in the Czech language to the respondents to understand the map topic. Even though it was not possible to cover all unusual visualisation styles nor all different regions, the Ref-Set covered the most frequent map types used in daily life while representing the huge diversity of the cartographic production. Such various set of sample maps was not found in any previous study focusing on map load evaluation.

All approaches, including their parameters developed in this study, are, of course, not the only potential solutions. Other image compression methods could be examined as well as various edge detection methods. This study aimed, though, to find which approach seems to be most effective in the estimation of graphic map load of various map styles as most other studies focused only on a single map type, cartographic method or map style. According to the study results, a significant correlation with user-perceived map load was found for IC1, IC2, IC3, AD1, AD2 and AD3 metrics while no significant correlation for the AD metric. Nevertheless, the ED approach seems to be more suitable for graphic map load evaluation than two other suggested approaches as the ED2 and ED3 metrics provided the best correlations with UP (0.77) and were not strongly affected by diversity in bit depth as the IC1 and IC2 metrics were. Therefore, the ED approach will be examined deeply in further research, where different filters and their behaviour will be examined concerning the graphic map load.

All metrics provided enough easy-to-use tools for quick evaluation of map load. All steps of the measurement were done manually using graphic user interface of the software, so calculating precise processing time is not possible as this is affected mostly by the agility of the evaluator. However, all maps from the Ref-Set were possible to be evaluated on an ordinary laptop (equipped with 2.9 GHz processor, 8GB RAM, Windows 64-bit operating system) in a short time. The most time-consuming operation, the edge detection, in the case of the largest map sample M17 (23.623 × 17.989 pixels), took 9 s. The histogram tool provided results immediately so the average brightness could be read. In addition, resaving files and their compression took only a few seconds. Examined values were noted into an Excel sheet with predefined Equations to compute map load values for all seven metrics. Therefore, the presented approaches could easily be applied on a variety of single-sheet maps.

## 5. Conclusions

In the study, three approaches for graphic map load were applied and compared with the user-perceived values achieved during user testing. For both the principles, the same Ref-Set of 50 various maps was used. The most promising results were obtained with ED metrics applying edge detection filter followed by measuring the average pixel brightness of the processed image map representation. Especially for metrics ED2 and ED3, high correlation coefficient 0.77 was registered. Therefore, edge detection is an approach, which is going to be examined more deeply in ongoing research.

Metrics based on average pixel darkness and image compression (AD, IC1, IC2 and IC3) were found fitting the user-perceived map load only for limited map styles. The AD metric was successfully evaluating maps with bright background and darker map symbols, typically topographic maps. Conversely, inverse map colouring significantly increased the measured values of graphic map load. Moreover, different levels of contrast do not allow to use this approach for comparison of various map styles. On the other hand, the image-compression metrics suffered from undervaluation of the map load values in the case of maps with the limited colour count used. Despite these limitations, the correlation was found between UP and all three metrics of the IC approach (IC1, IC2 and IC3).

Even though the ED metrics' results showed significantly better correlation with UP map load values, it should be considered that for other Ref-Set or respondent group correlations could differ slightly. Furthermore, the UP values of perceived map load relate to respondents' judgements, which may differ in time, with improving skills and other conditions. This supports the idea that designing a stable objective metric for analysing graphic map load is worth and may help cartographers to design more suitable maps.

**Author Contributions:** Conceptualization, Radek Barvir and Vit Vozenilek; methodology, Radek Barvir and Vit Vozenilek; software, Radek Barvir; validation, Vit Vozenilek; formal analysis, Radek Barvir and Vit Vozenilek; investigation, Radek Barvir; writing—original draft preparation, Radek Barvir; writing—review and editing, Vit Vozenilek; visualization, Radek Barvir; supervision, Vit Vozenilek; project administration and funding acquisition, Vit Vozenilek. All authors have read and agreed to the published version of the manuscript.

**Funding:** The paper was compiled within the Czech Scientific Foundation project No 18-05432S, Spatial synthesis based on advanced geocomputation methods and the project "Advanced Application of Geospatial Technologies for Spatial Analysis, Modelling, and Visualization of the Phenomena of the Real World" (IGA_PrF_2020_027) with the support of the Internal Grant Agency of Palacky University Olomouc.

**Conflicts of Interest:** The authors declare no conflicts of interest.

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
