# Peer review of "Developing Versatile Graphic Map Load Metrics"

_ijgi, doi:10.3390/ijgi9120705_

Round 1

Reviewer 1 Report

Thank you for the opportunity to review this article. I like the subject of the article, but I have comments.

- The selection of a group of people for usibilty test is one of the basic assumptions. The authors only provide the number of respondents and the level of cartographic education. It is worth adding what was the proportion of people with and without experience. Take this fact into account in further considerations (lines 99-103).

- Lines 307-308 „It can be further examined how the results would differ if the respondent group was different.” - I would give up such words.

- Viewing the map for 18 seconds, from where such a time (lines 107-109). What theoretical basis does this have, please add here the literature that confirms the practical use of exactly such time. Why was the unlimited time and the reduction of map variants not used?

- On what basis the authors assumed that exactly 25 maps, i.e. exactly half, will not lead to getting bored and inattention of the respondent (lines 113-116).

- I also think that giving the distribution of the data would improve the understanding of the choice of the correlation test. It seems to me (I can be wrong) that the set of obtained data should be tested by a group of nonparametric tests.

- I also think that the LSE parameter creates three groups that fit perfectly into the use of the Kruskal-Wallis test, which could enrich the analysis.

- p-value is missing in all statistical results. whether the results are statistically significant at all, so that any conclusions can be made.

- lines 132-137 - Please clarify the statements made. The conclusion that the authors use is very general and cannot be accepted as it form.

- Chapter 2.3 I hope that all maps were in the same resolution, because this element was not taken into account and it definitely affects the file size. Moreover, if all maps were in the same resolution, the method was not universal.

Reviewer 2 Report

An interesting article that tackles a rather peculiar problem concerning the cognitive science and map making. I found it sufficiently clear and well documented.

Reviewer 3 Report

An interesting and relevant paper overall.
I think there are two major things that should be addressed prior to publication:
1) What is the sensitivity of these map load metrics to the bit depth used for scanning the maps? An exemplary sensitivity analysis would be useful.
2) What is the potential of these metrics for large-scale complexity analysis of whole map archives / map collections? Some information on the computational cost of the proposed methods would be useful.

Major comments:

1) The presented experiments were conducted on scanned maps. How does the bitdepth used for scanning affect the results?
It seems the chosen image processing methods (brightness analysis, edge detection) are relatively insensitive to this, but
what about the compression based method? For example, if a map was scanned using different levels of bit depth, the scanned image
would have different levels of "noise", due to the paper granularity etc.
Would the image compression based metrics yield different load measures for the same map, scanned at different bit depths?
Please discuss this potential issue in the paper, and, if possible, add a small experiment (i.e., a sensitivity analysis) of the load metrics
to scanning bit depth. The authors could also use a color reduction method (e.g., k-means / mean shift clustering) on selected
map images to "simulate" lower levels of bit depth, rather than actually perform the scanning process.

2) These methods are very useful for several purposes.
As the authors mention, these metrics allow to assess the map complexity, and enable the map creator to assess the complexity, and
reduce complexity, if necessary.
Also, besides the use of these methods in mapmaking itself, the presented metrics are useful to analyze existing, digitally available (historical) map archives
(e.g., USGS topographic map archive, Sanborn fire insurance map archive, UK Ordnance survey, Swiss Siegfried maps, etc...)
Thus, the presented methods contribute to "image mining" or "geographic information retrieval" and possibly allow for quantifying the complexity
of whole archives / collections of maps. I suggest that the authors add these thoughts to the introduction or conclusion section.
See e.g.,
Uhl, J. H., Leyk, S., Chiang, Y. Y., Duan, W., & Knoblock, C. A. (2018). Map archive mining: visual-analytical approaches to explore large historical map collections.
ISPRS international journal of geo-information, 7(4), 148.

Petitpierre, Rémi, ‘Neural Networks for Semantic Segmentation of Historical City Maps: Cross-Cultural Performance and the Impact of Figurative Diversity’ (EPFL, Lausanne, 2020)  dx.doi.org/10.13140/RG.2.2.10973.64484

3) Apologies if I missed it, but is there any information on the processing times to obtain the different metrics?
I assume that the ED metric is the computationally most expensive metric. What are the processing times for the three methods?
In particular for the application of these metrics to large map archives, computational cost is critical.
It would be great to include some information / reflections on this issue in the discussion.

Minor comments:
Abstract line 12-14.
This applies not only to these specific cases, but to any map.
I would write:
"Generally, map design needs to enable the user to quickly, comprehensively, and intuitively obtain the relevant spatial information from a map."
or something like that.

Chapter 1:
The introduction nicely summarizes existing work and approaches.

Line 145: the "AD" in "LAD" should be subscript, to be consistent.
Line 182: What does "EDD" stand for? Please make sure that all acronyms are defined in the manuscript.
Line 187: I am confused about the metrics "ED2" and "ED3". Aren't L_ED2 and L_ED3 the metrics?
Line 209, 212 etc.: The "UP" in LUP needs to be set to subscript?
Line 340: Also here, I would speak of L_ED2, L_ED3, rather than ED2, ED3. Please make sure this is consistent thoughout the manuscript.
The less acronyms are used, the better the reader can follow.

Round 2

Reviewer 1 Report

Thanks again for the opportunity to review this article. I am fully satisfied with the changes made to the article.

Reviewer 3 Report

The paper has improved and all revision requests have been complied with. In my opinion, this manuscript is ready for publication.